# Biologically Oriented Hybrids of Indole and Hydantoin Derivatives

**DOI:** 10.3390/molecules28020602

**Published:** 2023-01-06

**Authors:** Konstantin A. Kochetkov, Olga N. Gorunova, Natalia A. Bystrova

**Affiliations:** A. N. Nesmeyanov Institute of Organoelement Compounds, Russian Academy of Sciences, 28 ul. Vavilova, 119991 Moscow, Russia

**Keywords:** indoles-imidazolidin-2-on hybrids, amidoalkylation, *bis*-heterocycles, growth-regulating activity

## Abstract

Indoles and hydantoins are important heterocycles scaffolds which present in numerous bioactive compounds which possess various biological activities. Moreover, they are essential building blocks in organic synthesis, particularly for the preparation of important hybrid molecules. The series of hybrid compounds containing indoles and imidazolidin-2-one moiety with direct C–C bond were synthesized using an amidoalkylation one-pot reaction. All compounds were investigated as a growth regulator for germination, growth and development of wheat seeds (*Triticum aestivum* L). Their effect on drought resistance at very low concentrations (4 × 10^−5^ M) was evaluated. The study highlighted identified the leading compounds, **3**a and **3**e, with higher growth-regulating activity than the indole-auxin analogues.

## 1. Introduction

The hydantoin or glycolylurea (imidazolidin-2,4-one) is an important nitrogen-containing heterocyclic moiety in which two nitrogen atoms are located in a five-membered ring. Hydantoin and its derivatives are common bioactive molecules, so synthetic and pharmacological interest in them is constantly growing [1,2,3,4]. In addition, neuroprotective [5], anticonvulsant [6], antibacterial [7,8], anti-inflammatory [5,9], and anti-cancer [10,11] properties of imidazolidin-2,4-one derivatives are known.

On the other hand, the indole skeleton represents a main structural motif in various bioactive natural products and pharmaceutically active compounds.

Its derivatives possess a broad spectrum of biological activities, including antimicrobial [12,13,14,15], antitubercular [16,17,18], anti-tumor [19,20,21,22,23], antiviral [17,24,25], antioxidant [12,21], anti-inflammatory [17,25,26,27], antifungal [28,29,30], antimalarial [17,31], antidepressants [17], antidiabetic [25,26,32], аnti-Alzheimer’s [26,33], and antibacterial [26,34] properties.

It has previously been shown that hybrid molecules containing two or more heterocyclic fragments in one structure exhibit synergistic or additive pharmacological activity compared to compounds with each individual pharmacophore [35,36]. Thus, in bioactive molecules, it is possible to change the pKa, to develop conformational hindrances and the regulation of the hydrophilic-hydrophobic balance, and to increase lipophilicity, and all these parameters can be varied both individually and in various combinations [37,38,39].

In the last decade, interest in biologically active hybrids based on indole and various azoles has increased. For example, indole-functionalized isoxazoles [40], indole-imidazole [9], and indole-imidazolone [41] hybrids have anti-inflammatory properties, indole-oxadiazole hybrids are potential antidiabetic agents [42], and indole-chalcone linked 1,2,3-triazole hybrids exhibit antimicrobial activity [43]. A whole series of indole hybrids with azoles, such as pyrazole [44], thiazolidinone [45], isoxazoles [46], benzimidazole [47], and 1,2,4-triazole [48] showed potential antitumor and cytotoxic activity.

Indole derivatives are also known to be widely used in agriculture. It is known that derivatives of indole auxins are phytohormones. They affect the growth and development of plants and increase their resistance to the environment [49,50,51,52,53]. For example, indolylacetic (IAA) and indolebutyric (IBA) acids act as growth regulators, stimulate root formation [54,55,56], flower development, and fruit growth [57], and help plants overcome stress [58]. Tryptophan improves plant resistance to Cd, reduces its accumulation in the biomass [59], and increases yields in nutrient-deficient soils [53]. Indole derivatives with fungicidal activity, such as streptochlorin, are well-known [28,30].

At the same time, some hydantoin derivatives also have applications in the agrochemical area as bactericides, fungicides, and herbicides [60,61,62,63,64]. However, according to our information, the usage of imidazolidinone-based compounds as plant growth regulation agents has not yet been explored. In a preliminary report, we described the synthesis and study of the cytotoxic and anti-inflammatory properties of the series binary 3-/2- (2-oxoimidazolidinyl-5)-indoles [65].

The aim of the work was the synthesis of biologically oriented hybrids of hydantoin and C2/C3–substituted indoles linked by a direct C-C bond and the study of the derivatives of (2–oxoimidozolidinyl-5)-indoles for growth-regulating activity.

## 2. Results and Discussion

### 2.1. Synthesis

Imidazolylindole derivatives with a direct C-C bond are relatively hard toreach. Conventional approaches to obtaining these hybrid compounds, as a rule, include multi-stage synthesis with intramolecular condensation of the intermediate product at the last stage [66]. At the same time, the *C*-amidoalkylation reaction is a simple one-pot and effective method of introducing new fragments into the heterocycle molecule [67]. Amidoalkylated reagent 1-phenylimidazolidin-2-one **1** was obtained from the corresponding hydantoin derivative using the known procedure [68]. The synthetic pathway used in the preparation of novel hydantoin derivatives containing indole moiety with a direct-linked C-C bond is shown in Figure 1. Amidoalkylation was carried out under Lewis acid catalysis. Performing the reaction on indole **2**a and *N*-methylindole **2**d as well as the **2**-substituted indoles **2**b,c,e,f in THF in the presence of catalytic amounts of boron trifluoride ether provided satisfactory results to afford the desired indole-imidazolidine-2-on hybrids **3**a–f in 37–54% yields. Similarly, 2-(2-oxoimidazolidinyl-5)-indole (**3**g) was synthesized, although with a lower yield.

The usage of other catalysts (diethylaminoethylcellulose, SiO_2_, Al_2_O_3_), organic bases (Et_3_N, DIPEA), and solvents (DCE, chloroform, and dioxane) leads to a significant decrease in the product yield. Products of the **2**,**3**-rearrangement of Wagner–Meerwein, characteristic of indole derivatives in an acidic medium and present in reaction mixtures of indolylpyrazolidines [69], were not detected under these conditions. It should be noted that the sulfur-containing analogue of reagent **1** does not interact with indoles [70,71].

The structure of the synthesized indole-imidazolidine-2-on hybrids **3**a–g was confirmed using ^1^H NMR spectroscopy, with signal assignments based on the gCOSY techniques. In the aliphatic part of the ^1^H NMR spectra of the heterocyclic hybrids **3**, proton signals of the starting 1-phenylimidazolidine-2-one is preserved. For all compounds **3**, there is no signal in the ^1^H NMR spectra at δ_H_ 6.42 corresponding to the OH group proton of the starting reagent **1**, which indicates the formation of a C–C bond in the C(5’) position of imidazolidin-2-on derivatives. Comparative analysis of the amidoalkylation products of isomeric 2- and 3-methylindoles (such as, for example, **3**b and **3**g) showed that they have different physico-chemical properties, and their ^1^H NMR spectra do not contain similar signals. In all ^1^H NMR spectra of compounds **3**a–f, compared to the spectra of starting indoles **2**a–f, the signal for the C(3)H proton of the indole core is absent, which confirms the direction of the amidoalkylation of the indoles at position C(3), if it is free. The ^1^H NMR spectra of imidazolidinylindoles **3**a–c, **3**g obtained from the *N*-unsubstituted indoles exhibit a characteristic broadened signal for the proton of the NH group at δ 8.46 for derivative **3**g and at δ_H_ 10.87–11.32 for compounds **3**a–c, which excludes the nucleophilic attack at position C(1) of the indole. The assignment of carbon atom signals in the ^13^C NMR spectra was carried out using two-dimensional heteronuclear correlation experiments, as well as by comparing them with the spectra of a wide range of substituted indoles. [72,73]. The structures of compounds **3**a and **3**f were established earlier in X-ray diffraction studies [65].

Tests on the cells of mouse microglia of the BV-2 line (CVCL_0182) and human neuroblastoma of the SH-SY5Y line (ATCCCRL-2266) showed that indole-imidazolidine hybrids **3**a–f manifested cytotoxicity and an anti-inflammatory effect [65].

### 2.2. Investigation of the Influence of Hydantoin-Indole Hybrids on Wheat Seed Germination

As model compounds, we have chosen some synthesized derivatives of 3-(2-oxoimidazolidine-5-yl)indole **3** to study growth-regulating activity. Known auxins [49,50,51,52,53]—derivatives of indole, namely indolylacetic (IAA) and indolebutyric (IBA) acids, as well as L-tryptophan (Trp)—were selected as compounds for comparison.

It is known that the concentration of 10^−7^ M is typical for these indole-auxins in agriculture [74]. However, modern research has shown that growth regulators with individual concentrations are selected for each crop (Appendix A). According to these statistics, the concentration of PGR 4 ×10^−5^ M that we used is acceptable for wheat seeds. Seeds of spring wheat (*Triticum aestivum* L.) of the “Darya^®^” variety, crop 2020 (No. 9705798), were selected for research [75]. The results of the research are presented in Table 1.

Seed germination is a key stage in the plant’s life cycle. The first 24 h of the experiment were conducted in the dark to simulate the development of seeds in the natural environment, in the soil. The seed germination potential (Gp) was calculated after 24 h, according to Appendix A. The root (1–3 mm) is the evaluation criterion Gp, as it penetrates through the seed membranes [76]. The results for germination potential and germination are presented on the Figure 1.

All the tested compounds **3**a–g have shown a good growth-regulating ability. Hybrid compound **3**e-treated seeds have the highest Gp (83%) compared to the control (77%). Germination (G) of the seeds was calculated on the seventh day using Appendix A, according to the recommendations of the International Seed Testing Association [77]. Seeds treated with compounds **3**a,c,e showed higher germination results than the control sample (89%). The maximum value was for the seeds treated with compound **3**a (95%).

Root system architecture plays an important role in plant development [78]. It is generally accepted that a deeper and more branched root system can give advantages when growing crops in difficult conditions [79]. It is known that the root system consists of two types of roots: the primary root (PR) and lateral roots (LRs) (Appendix A).

The study of the root length and the shoot height was carried out on the seventh day after the start of the experiment (Appendix A). The root system of the control sample does not have a primary root; there are only four roots, and their average length is presented in Table 1. All other samples have a root system with primary and lateral roots. Only sample 3e began to develop root hairs on the lateral roots by the seventh day. Compound **3**a showed the best results: the main root was larger than 10 cm, and the average size of the lateral roots was larger than 7 cm. The results of shoot height studies were statistically insignificant. The seeds treated with the tested compounds showed results comparable to the control (about 13 cm). Thus, the use of hybrid compounds leads to the accelerated growth and branching of the root system by the seventh day.

So, hybrids of the 3-substituted indoles (**3**a–c) demonstrated greater activity compared to the 2-substituted analog **3**g. An increase in the volume of the substituent in the second position of the indole ring (for example, **3**a–**3**c) does not significantly affect the growth-regulating properties. At the same time, 1-methyl derivatives (**3**d,f) had less of an effect on the growth and development of wheat seeds.

Drought is an environmental stress factor for plants. A lack of water can provoke the restriction of photosynthesis, drying of biomass, and reduction of shoot height. The relative water content (RWC) of leaf is a reliable and simple way to assess the water status of a plant. It is used to describe the state of water in a plant at a particular point in time. Higher RWC readings are an indication of drought tolerance [80]. The RWC was calculated using Formula (S3). The results of the research are presented in Table 2.

The relative water content in the RWC was determined at 72, 96, and 120 h (Figure 2). after the last watering, as we wanted to understand how the RWC of the leaves changes when watering stops.

Throughout the 72 h after the last watering, all the plants retained their turgor, and after 96 h, the beginning of the wilting process was observed. The shoots had a minimum degree of wilting and a maximum value of 30.31% RWC when treated with compound **3**a compared to the control (24.38%). The shoots treated with compound **3**e (27.94%) and IAA (27.63%) had the same value. Thus, new hybrid compounds can be used to improve the drought tolerance of wheat plants.

## 3. Materials and Methods

### 3.1. Chemicals and Instruments

The ^1^ H NMR and ^13^C NMR spectra were recorded on Agilent 400-MR (400 MHz) and Bruker Avance-600 spectrometers (600 MHz) in CDCl_3_ or DMSO-d_6_ using tetramethylsilane (TMS) as internal standard. The chemical shifts were reported in *δ* scale, and constant *J* values are presented in Hz. IR spectra were recorded on a UR-20 instrument in Nujol and on an IR-200 Fourier-transform IR spectrometer (TermoNicolet, Waltham, MA, USA) with a resolution of 4 cm^−1^ (KBr pellets). Electrospray ionization (ESI) high-resolution mass spectra were recorded on a Bruker maXis instrument. Melting points were measured on an Electrothermal IA 9000 series device in glass capillaries. Elemental analysis was performed on a Carlo Erba device EA 1108CHNS-O. The TLC on Silufol UV-254 was used to follow the course of reactions. Compound purification was performed using short dry column or flash-chromatography on silica gel (60, Fluka, Honeywell Research Chemicals, Morris Plains, NJ, USA) [81]. Started indole derivatives were used as purchased from Sigma-Aldrich. The 5-Hydroxy-1-phenylimidazolidin-2-one, **1** was obtained according to the procedure [61]. Solvents were purified by standard methods.

### 3.2. Reaction of Indoles **2** with Phenylimidazolidin-2-One, **1** (General Procedure)

The catalyst BF_3_•Et_2_O (16 mg, 0.1 mmol) was added to a mixture of 5-hydroxy-1-phenylimidazolidin-2-one **1** (1 mmol) and the corresponding indole **2** (1 mmol) in anhydrous THF (10 mL) with stirring. The reaction mixture was stirred at room temperature (1–6 h, TLC control) and filtered through a short layer of SiO_2_, the residue was washed with CHCl_3_ on the filter, and the solvent was evaporated in vacuo. Diethyl ether (~5 mL) was added to the oily residue and triturated to form a fine crystalline precipitate. An additional recrystallization of this precipitate from EtOH gave the corresponding indolylimidazolidinone **3**.

#### 3.2.1. 5-(1H-Indol-3-yl)-1-Phenylimidazolidin-2-One (**3**a)

The yield was 50%, white crystals. M.p. 244 °C. IR (KBr, n/cm^−1^): 1680 (C = O), 3350 (NH). ^1^H NMR (DMSO-d_6_): 3.36 (m, 1H, C(4’)H^a^), 3.86 (td, 1H, J = 9.2 Hz, J = 3.2 Hz, C(4’)H^b^), 5.73 (dd, 1H, J = 9.2 Hz, J = 6.2 Hz, C(5’)H), 6.87 (t, 1H, J = 7.3 Hz, *p*-CH_Ph_), 6.95 (s, 1H, N(3’)H); 6.99 (t, 1H, J = 7.5 Hz, C(5)H), 7.07 (t, 1H, J = 7.5 Hz, C(6)H), 7.14 (t, 2H, J = 7.9 Hz, *m*-CH_Ph_), 7.31 (s, 1H, C(2)H), 7.33 (d, 1H, J = 7.4 Hz, C(7)H), 7.47 (d, 2H, J = 7.9 Hz, *o*-CH_Ph_), 7.62 (d, 1H, J = 7.6 Hz, C(4)H), 10.87 (s, 1H, N(1)H). ^13^C NMR (DMSO-d_6_): 45.8 C(4’), 53.86 C(5’), 112.25 C(7), 114.58 C(3), 119.15 C(5), 119.34 C(4), 120.88 (2 C, *o*-CPh), 121.7 C(6), 122.7 *p*-CPh, 124.6 C(2), 125.5 C(**3**a), 128.4 (2 C, *m*-CPh), 137.3 C(7a), 140.2 C(1)Ph, 159.6 (C = O). Found (%): C, 71.41; H 5.46; N 14.23. C_17_H_15_N_3_O•0.5H_2_O. Calculated (%): C, 71.31; H, 5.63; N,14.68.

#### 3.2.2. 5-(2-Methyl-1H-Indol-3-yl)-1-Phenylimidazolidin-2-One (**3**b)

The yield was 48%, a white powder. M.p. 275 °C. IR (KBr, n/cm–1): 1681 (C = O), 3240 (NH). ^1^H NMR (DMSO-d_6_): 2.41 (s, 3H, C(2)CH_3_), 3.33 (m, 1H, C(4’)H^a^), 3.79 (t, 1H, J = 9.2 Hz, C(4’)H^b^), 5.72 (dd, 1H, J = 9.2 Hz, J = 7 Hz, C(5’)H), 6.83 (t, 1H, J = 7.3 Hz, *p*-CHPh), 6.91 and 6.95 (both m, 1H each, C(5)H, C(6)H), 7.09 (s, 1H, N(3’)H), 7.12 (t, 2H, J = 7.8 Hz, *m*-CHPh), 7.17 (d, 1H, J = 7.8 Hz, C(7)H), 7.32 (d, 2H, J = 8 Hz, o-CHPh), 7.47 (d, 1H, J = 7.9 Hz, C(4)H), 10.88 (s, 1H, N(1)H). ^13^C NMR (DMSO-d_6_): 11.7 C(2)CH_3_, 44.9 C(4’), 52.5 C(5’), 109.2 C(3), 111.1 C(7), 118.3 C(4), 119.1 C(5), 120.8 (2 C, *o*-CPh), 120.9 C(6), 122.8 *p*-C(Ph), 126.4 C(**3**a), 128.4 (2 C, m-CPh), 133.8 C(2), 135,6 C(7a), 139.7 C(1)Ph, 159.5 (C = O). Found (%): C, 74.51; H, 6.27; N, 14.32. C_18_H_17_N_3_O. Calculated (%): C, 74.20; H, 5.88; N 14.32.

#### 3.2.3. 5-(2-*p*-Tolyl-1H-Indol-3-yl)-1-Phenylimidazolidin-2-One (**3**c)

The yield was 54%, white crystals. M.p. 317 °C. IR (KBr, n/cm^−1^):1684 (C = O), 3200–3300 (NH). ^1^H NMR(DMSO-d_6_): 2.42 (s, 3H, Tol-CH_3_), 3.56, 4.03 (both t, 1H, J = 8 Hz, C(4’)H^a^, C(4’)H^b^), 5.62 (t, 1H, J= 8 Hz, C(5’)H), 6.78 (t, 1H, J = 8 Hz, *p*-CHPh), 6.91–7.08 (m, 6H, C(5)H, C(6)H, *o*-CHPh, *m*-CHPh), 7.19 (s, 1H, N(3’)H), 7.31 (d, 1H, J = 7.4 Hz, C(7)H), 7.40 and 7.44 (both d, 2H each, J = 7.4 Hz, *o*-C*H*Tol and *m*-C*H*Tol), 7.59 (d, 1H, J = 7.6 Hz, C(4)H), 11.32 (s, 1H, N(1)H). ^13^C NMR (DMSO-d_6_): 21.3 (Tol-CH_3_), 44.9 C(4’), 52.9 C(5’), 109.8 C(3), 112.0 C(7), 119.5 C(4), 119.7 C(5), 120.3 (2 C, *o*-CPh), 122.0 C(6), 122.6 *p*-CPh, 126.2 C(**3**a), 128.3 (2 C, *m*-CPh), 129.0 and 129.9 each (2 C, *o*- and *m*-CTol), 136.6 C(2), 137.6 C(**7**a), 138.1 C(1)Tol, 139.7 C(1)Ph, 159.4 (C = O). Found (%): C, 74.27; H, 5.69; N, 10.70. Calculated for C_24_H_21_N_3_O•H_2_O: C, 74.78; H, 6.01; N, 10.90.

#### 3.2.4. 5-(1-Methy-1H-Indol-3-yl)-1-Phenylimidazolidin-2-One (**3**d)

The yield was 49%, white crystals. M.p. 233 °C. IR (KBr, n/cm^−1^): 1695 (C = O), 3000–200 (NH). ^1^H NMR (DMSO-d_6_): 3.3 (dd, 1H, J = 8.8 Hz, C(4’)H^a^), 3.68 (s, 3H, N(1)CH_3_), 3.84 (t, 1H, J = 9.1 Hz, C(4’)H^b^), 5.72 (dd, 1H, J = 6.1 Hz, J = 9.1 Hz, C(5’)H), 6.87 (t, 1H, J = 7.4 Hz, *p*-CHPh), 7.02 (dd, 1H, J = 8 Hz, J =7 Hz, C(5)H), 7.09–7.11 (m, 2H, C(6)H, N(3’)H), 7.15 (m, 2H, *m*-CHPh), 7.33 (s, 1H, C(2)H), 7.36 (d, 1H, J = 8 Hz, C(7)H), 7.46 (d, 2H, J = 8 Hz, o-CHPh), 7.62 (d, 1H, J = 8 Hz, C(4)H). ^13^C NMR (DMSO-d_6_): 32.8 N(1)CH_3_, 45.7 C(4’), 53.2 C(5’), 110.5 C(7), 113.5 C(3), 119.3 C(4), 119.5 C(5), 120.6 (2 C, *o*-CPh), 121.9 C(6), 122.7 *p*-C(Ph), 125.6 C(**3**a), 128.5 (2 C, *m*-CPh), 128.8 C(2), 137.5 C(7a), 139.9 C(1)Ph, 159.5 (C = O). Found (%): C, 71.77; H, 5.72; N, 13.63. C_18_H_17_N_3_O•0.5H_2_O. Calculated (%): C, 71.98; H,6.04; N, 13.99.

#### 3.2.5. 5-(1,2-Dimethyl-1H-Indol-3-yl)-1-Phenylimidazolidin-2-One (**3**e)

The yield was 37%, white crystals. M.p. 278 °C.IR (KBr, n/cm^−1^):1699 (C = O), 3100–3200 (NH). ^1^HNMR (DMSO-d_6_): 2.45 (s, 3H, C(2)CH_3_), 3.32 (t, 1H, J = 8.3 Hz, C(4’)H^a^), 3.48 (br.s, 1H, N(3’)H), 3.58 (s, 3H, N(1)CH_3_), 3.8 (t, 1H, J = 9.2 Hz, C(4’)H^b^), 5.78 (t, 1H, J = 8.3 Hz, C(5’)H), 6.85 (t, 1H, J = 7.3 Hz, *p*-CHPh), 6.95 (t, 1H, J = 7.3 Hz, C(5)H), 7.03 (t, 1H, J = 7.3 Hz, C(6)H), 7.12 (t, 2H, J = 7.3 Hz, *m*-CHPh), 7.31 (d, 1H, J = 8 Hz, C(7)H), 7.33 (d, 2H, J = 8 Hz, *o*-CHPh), 7.52 (br.m, 1H, C(4)H). ^13^C NMR (DMSO-d_6_): 10.3 C(2)CH_3_, 29.7 N(1)-CH_3_, 45.0 C(4’), 52.8 C(5’), 109.3 C(3), 109.8 C(7), 118.4 C(4), 119.4 C(5), 120.9 (3 C, *o*-CPh and C(6)), 122.9 *p*-C(Ph), 128.8 (2 C, *m*-CPh), 135.5 C(2), 137.0 C(7a), 139.7 C(1)Ph, 159.9 (C = O). Found (%): C, 73.18; H, 6.07; N, 13.32. C_19_H_19_N_3_O•0.5H_2_O. Calculated (%): C, 72.59; H, 6.41; N, 13.37.

#### 3.2.6. 5-(1-Methyl-2-*p*-Tolyl-1H-Indol-3-yl)-1-Phenylimidazolidin-2-One (**3**f)

The yield was 52%, white crystals. M.p. 183 °C. IR (KBr, n/cm^−1^): 1701 (C = O) 3000–3200 (NH). ^1^H NMR (CDCl_3_): 2.45 (s, 3H, Tol-CH_3_), 3.49 (s, 3H, N(1)-CH_3_), 3.87 (m, 2H, C(4’)H_2_), 5.32 (br.s, 1H, N(3’)H), 5.4 (t, 1H, J = 8.8 Hz, C(5’)H), 6.94 (t, 1H, J = 6.6 Hz, *p*-CH_Ph_), 7.08–7.17 (m, 7H (2 + 2+2 + 1), *m*-CHPh, *o*-CHTol, *m*-CHTol, C(5)H), 7.24 (t, 1H, J = 7.6 Hz, C(6)H), 7.3 (d, 1H, J = 7.6 Hz, C(7)H), 7.36 (d, 2H, J = 7.6 Hz, *o*-CHPh), 7.82 (d, 1H, J =8 Hz, C(4)H). ^13^C NMR (CDCl_3_): 30.6 N(1)CH_3_, 45.3 C(4’), 54.5 C(5’), 109.6 C(3), 110.2 C(7), 119.8 C(4), 120.0 C(5), 122.0 C(6), 122.4 *p*-CPh, 125.1 C(**3**a), 128.2 (2 C, *m*-CPh), 129.3 and 130.4 (2 C, each *o*- and m-CTol), 137.4 C(7a), 138.4 (C2), 138.9 C(1)Tol, 140.1 C(1)Ph, 160.2 (C = O). Found (%): C, 78.44; H, 6.02; N, 10.65. C_25_H_23_N_3_O. Calculated (%): C, 78.71; H, 6.08; N,11.02.

#### 3.2.7. 5-(3-Methyl-1H-Indol-2-yl)-1-Phenylimidazolidin-2-One (**3**g)

The yield was 34%, white crystals. M.p. 201 °C. ^1^H NMR (DMSO-d_6_): 2.28 (s, 3H, C(3)CH_3_), 3.83 (t, 1H, J = 8.2 Hz, C(4’)H^a^), 4.23 (t, 1H, J = 9.2 Hz, C(4’)H^b^), 5.21 (dd, 1H, J = 7.8 Hz, J = 9.5 Hz, C(5’)H), 7.00 (m, 2H, C(6)H, *p*-CHPh), 7.08 (m, 1H, J = 7.5 Hz, C(5)H), 7.32 (m, 3H, *m*-CHPh, C(4)H), 7.46 (d, 1H, J = 8 Hz, C(7)H), 7.64 (d, 2H, J = 8.5 Hz, *o*-CHPh), 7.52 (s, 1H, N(1)H), 11.05 (s, 1H, N(1′)H). ^13^C NMR (DMSO-d_6_): 8.7 C(2)CH_3_, 44.9 C(4’), 51.1 C(5’), 107.9 C(3), 111.7 C(7), 117.6 C(5), 118.6 C(4), 117.6 (2 C, *o*-CPh), 121. 8 C(6), 122.1 C(6), 128.8 C(**3**a), 129.0 (2 C, m-CPh), 134.1 C(2), 136.3 C(**7**a), 141.1 C(1)Ph, 158.6 (C = O). MS: m/z + [M + H] 292.3531. Calculated for C_18_H_18_N_3_O 292.3551.

### 3.3. Investigation of The Growth-Regulating Activity of Compounds **3**

Wheat seeds (*Triticum aestivum* L.) of the “Darya^®^” variety, crop 2020, provided by LLC “Zhito,” Oktyabrsky district, Ryazan, Ryazan region, Russia 54.609836° S.w., 39.80188° V.D. were used. Four independent series of experiments using identical cameras with phyto-LED UFO lighting-79–01-00 with a wavelength of Red 615/Blu 457 nm with an intensity of at least 250 lux were carried out. The illumination of the samples is 12/12 h. The relative humidity of the air was 50 ± 2%. The temperature was 20 ± 2 °C. The duration of the experiment was 7 days. Fifty (50) pieces of dry sterilized seeds were placed on filter paper in rectangular Petri dishes 75 × 85 (mm) and treated by spraying with the studied compounds. Wheat grains were treated with 0.335 ± 0.003 mL of compound solutions. Statistical analysis was carried out using Microsoft Excel and STATISTICA 13.3 TRIAL programs (StatSoft Russia). ANOVA analysis of variance was performed to compare the data. Differences from the control (water) at *p* ≤ 0.05 were considered significant.

## 4. Conclusions

In summary, we have extended the efficient chemical strategy for the synthesis of asymmetrically hybrid compounds into products containing indoles and imidazolidin-2-one moiety with a direct C–C bond. Amidoalkylation has been shown to be an efficient one-pot approach to the preparation of these hard-to-reach compounds. For all the tested substances, the indicators of seed germination potential (Gp), germination (G), and relative water content (RWC) were obtained and analyzed. The compounds showed high growth-regulating activity on the wheat seeds in comparison with known indole-auxin analogues, such as indolylacetic and indolebutyric acids, and *L*-tryptophan. Their positive effect on drought resistance was also established even at very low concentrations of the substances. New hybrid compounds can be used to increase the resistance of wheat (*Triticum aestivum* L.) plants to negative environmental factors.

## Data Availability

Not applicable.

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
