# Peer review of "Biologically Oriented Hybrids of Indole and Hydantoin Derivatives"

_molecules, 2023, doi:10.3390/molecules28020602_

Round 1

Reviewer 1 Report

1- in introduction part:

- Please add Hydantoin, or glycolylurea.

- Delete  in which two nitrogen atoms are located in a five-membered ring.

- You need to discuss the main synthetic route to some hybide idole/Hydantoin compounds.

2-  Four  compounds were prepared in very low yield: the authors can try another catalyst and reaction conditions to improve the yields.

3- Compounds 3a and 3f are known and published in 2020.

4- assignments needed for all 13C NMR data.

5- I revise only the chemistry part; since the biological part is out of my special

Author Response

Article Title: Biologically oriented hybrids of indole and hydantoin derivatives

Letter Subject: Manuscript ID: molecules-2115192

Letter:
Dear Editor,
thank you for your e-mail message  and for giving us a possibility to revise our manuscript. The referee’s reports were positive and useful and we would like to thank the referees for all suggestions to improve our manuscript. After careful reading of the comments we have corrected the article and supplemented it in accordance with the recommendations of the reviewers (changes are marked in red). Please find attached in electronic form a revised version of our manuscript in which all scientific concerns noted by the reviewer have been addressed. Below you will find a point-by-point response to the reviewers comments.

Response to Reviewer 1

Comments

Point 1:- in introduction part:

- Please add Hydantoin, or glycolylurea.

- Delete in which two nitrogen atoms are located in a five-membered ring.

 - You need to discuss the main synthetic route to some hybide idole/Hydantoin compounds.

Response 1:

Added

Deleted

Information was added in discussion of results section

Point 2-Four compounds were prepared in very low yield: the authors can try another catalyst and reaction conditions to improve the yields.

Response 2: We have studied different catalysts  and solvents  but the yield was below (see red text)

Point 3- Compounds 3a and 3f are known and published in 2020.

Response 3: Compounds 3a and 3f are known, but a detailed spectral analysis and study of the structure and biological properties as plant growth regulators have not been carried out

Point 4- assignments needed for all 13C NMR data.

Response 4: Text corrected and assignments for all 13C NMR data were added.

Point 5- I revise only the chemistry part; since the biological part is out of my special

Reviewer 2 Report

Gorunova and group reported the Synthesis and biological effect of indole and hydantoin hybrids. This paper can be published after addressing the following comments.

1.      Authors are suggested to provide 13C NMR and HRMS analysis in the results and discussion section of the synthesized compounds.

2.      Please provide the reaction time and temperature in scheme 1.

3.      Why authors did not evaluate all the synthesized compounds?

4.      Structures in Table 1 were merged into each other and some of them were wrongly drawn (IAA, IBA, and Trp). Please check the position of the double bond in the five-membered rings of the above-mentioned structures.

5.       Authors should provide the structure-activity relationship of the  synthesized compounds which helps the reader with the future design of the molecules.

6.       Please provide the supplementary file.

7.       Materials and methods for the biological section were missing.

8.       Please check the font size and style carefully. It should be the same throughout the manuscript.

9.      Authors mentioned in the conclusion that they have developed an efficient chemical strategy for the synthesis of 285 unsymmetrically hybrid compounds which is not convincing as they have already reported the synthesis in their previous paper  Mendeleev Communications Volume 30, Issue 3, 2020, 347

Author Response

Article Title: Biologically oriented hybrids of indole and hydantoin derivatives

Letter Subject: Manuscript ID: molecules-2115192

Letter:
Dear Editor,
thank you for your e-mail message  and for giving us a possibility to revise our manuscript. The referee’s reports were positive and useful and we would like to thank the referees for all suggestions to improve our manuscript. After careful reading of the comments we have corrected the article and supplemented it in accordance with the recommendations of the reviewers (changes are marked in red). Please find attached in electronic form a revised version of our manuscript in which all scientific concerns noted by the reviewer have been addressed. Below you will find a point-by-point response to the reviewers comments.

Response to Reviewer 2

Comments

Gorunova and group reported the Synthesis and biological effect of indole and hydantoin hybrids. This paper can be published after addressing the following comments.

Point 1. Authors are suggested to provide 13C NMR and HRMS analysis in the results and discussion section of the synthesized compounds.

Response 1: Information was added in the text.

Point 2. Please provide the reaction time and temperature in scheme 1.

Response 2: Information was added in scheme 1.

Point 3. Why authors did not evaluate all the synthesized compounds?

Response 3: Other substances showed intermediate activity results and at the request of the reviewer we have now included these data for the compound 3b.

Point 4 Structures in Table 1 were merged into each other and some of them were wrongly drawn (IAA, IBA, and Trp). Please check the position of the double bond in the five-membered rings of the above-mentioned structures.

Response 4: Structures were corrected

Point 5. Authors should provide the structure-activity relationship of the synthesized compounds which helps the reader with the future design of the molecules.

Response 5: Information was added in the text

Point 6. Please provide the supplementary file.

Response 6: Information was added early and additionally repeated

Point 7. Materials and methods for the biological section were missing.

Response 7: Information was added now

Point 8. Please check the font size and style carefully. It should be the same throughout the manuscript.

 Response 8: Text corrected

Point 9. Authors mentioned in the conclusion that they have developed an efficient chemical strategy for the synthesis of 285 unsymmetrically hybrid compounds which is not convincing as they have already reported the synthesis in their previous paper Mendeleev Communications Volume 30, Issue 3, 2020, 347

Response 9.  The authors extended the previously proposed chemical strategy to the synthesis of other unsymmetrical hybrid compounds, and the number 285 erroneously appeared in the article template as a text line number.

Reviewer 3 Report

Reviewers Comments (Manuscript ID: molecules-2115192) 

The manuscript by Dr. Kochetkov and co-workers reported the “Biologically oriented hybrids of indole and hydantoin derivatives.” In current article, author did synthesize eight-known indole and hydantoin derivatives and studied their growth regulator properties for germination, growth and development of wheat seeds and which resulted some of the derivatives with improved growth regulating activity.

This reviewer is recommending this manuscript for the publication in molecules after minor revision. 

1.     Since this article is based on growth regulating activity so author should elaborate introduction part with growth regulating or agrochemical rather than pharmacological activity of indole and hydantoin.

2.      Figure 3 is repetition of scheme 1.

3.     Reaction conditions are missing in the scheme 1.

4.     Page 3, 1H, 13C{1H} and NMR spectroscopy can be changed as NMR (1H and 13C) spectroscopy.

5.     Last para on page 3, is not making any connection and also it looks like copied from ref 59.

6.     Page 4, first paragraph not require to be in bold font.

7.     In table 1, some of the structures are overlapping each other.

8.     Formatting of manuscript need to be rechecked before resubmitting the revision.

Author Response

Article Title: Biologically oriented hybrids of indole and hydantoin derivatives

Letter Subject: Manuscript ID: molecules-2115192

Letter:
Dear Editor,
thank you for your e-mail message  and for giving us a possibility to revise our manuscript. The referee’s reports were positive and useful and we would like to thank the referees for all suggestions to improve our manuscript. After careful reading of the comments we have corrected the article and supplemented it in accordance with the recommendations of the reviewers (changes are marked in red). Please find attached in electronic form a revised version of our manuscript in which all scientific concerns noted by the reviewer have been addressed. Below you will find a point-by-point response to the reviewers comments.

Response to Reviewer 3

Comments

The manuscript by Dr. Kochetkov and co-workers reported the “Biologically oriented hybrids of indole and hydantoin derivatives.” In current article, author did synthesize eight-known indole and hydantoin derivatives and studied their growth regulator properties for germination, growth and development of wheat seeds and which resulted some of the derivatives with improved growth regulating activity.

This reviewer is recommending this manuscript for the publication in molecules after minor revision. 

Point 1.     Since this article is based on growth regulating activity so author should elaborate introduction part with growth regulating or agrochemical rather than pharmacological activity of indole and hydantoin.

Response 1: Recent data, references and corresponding text on the agrochemical use of hydantoins and indoles were added

Point 2.      Figure 3 is repetition of scheme 1.

Response 2: Figures 3 is excluded from the manuscript

Point 3.     Reaction conditions are missing in the scheme 1.

Response 3: Information was added in scheme 1.

Point 4.     Page 3, 1H, 13C{1H} and NMR spectroscopy can be changed as NMR (1H and 13C) spectroscopy.

Response 4: Text was corrected

Point 5.     Last para on page 3, is not making any connection and also it looks like copied from ref 59.

Response 5: Last para on page 3 is now excluded

Point 6.     Page 4, first paragraph not require to be in bold font.

Response 6: Text was corrected

Point 7.     In table 1, some of the structures are overlapping each other.

Response 7: The structures were corrected

Point 8.     Formatting of manuscript need to be rechecked before resubmitting the revision.

Response 8: Formatting of manuscript was rechecked

Round 2

Reviewer 1 Report

Dear 

Reviewer 2 Report

The authors have addressed all the comments and the manuscript can be accepted for publication.